# Machine Learning-Based Radiomic Features on Pre-Ablation MRI as Predictors of Pathologic Response in Patients with Hepatocellular Carcinoma Who Underwent Hepatic Transplant

**DOI:** 10.3390/cancers15072058

**Published:** 2023-03-30

**Authors:** Azadeh Tabari, Brian D’Amore, Meredith Cox, Sebastian Brito, Michael S. Gee, Eric Wehrenberg-Klee, Raul N. Uppot, Dania Daye

**Affiliations:** 1Department of Radiology, Massachusetts General Hospital, Harvard Medical School, 55 Fruit Street, Boston, MA 02114, USA; 2Department of Radiology, Baylor College of Medicine, Houston, TX 77030, USA

**Keywords:** hepatocellular carcinoma, machine learning, liver transplant, radiofrequency ablation, microwave ablation, risk prediction

## Abstract

**Simple Summary:**

Early-stage hepatocellular carcinoma (HCC) is best managed by curative treatment, which includes resection, ablation, or transplantation. Complete pathologic tumor response to ablation is below the resolution of standard imaging. We investigated the role of pre-ablation tumor radiomics in predicting pathologic treatment response in patients with early-stage HCC who underwent liver transplantation. Using data collected from 2005–2015, we identified 97 patients who had a contrast-enhanced MRI within 3 months prior to ablation therapy and underwent liver transplantation. A random forest model was developed based on top radiomic and clinical features. The random forest model included two radiomic features (diagnostics maximum and first-order maximum) and four clinical features (pre-procedure creatinine, pre-procedure albumin, age, and gender) achieving an AUC of 0.83, a sensitivity of 82%, a specificity of 67%, a PPV of 69%, and an NPV of 80%. Pre-ablation MRI radiomics could act as a valuable imaging biomarker for the prediction of tumor pathologic response in patients with HCC.

**Abstract:**

Background: The aim was to investigate the role of pre-ablation tumor radiomics in predicting pathologic treatment response in patients with early-stage hepatocellular carcinoma (HCC) who underwent liver transplant. Methods: Using data collected from 2005–2015, we included adult patients who (1) had a contrast-enhanced MRI within 3 months prior to ablation therapy and (2) underwent liver transplantation. Demographics were obtained for each patient. The treated hepatic tumor volume was manually segmented on the arterial phase T1 MRI images. A vector with 112 radiomic features (shape, first-order, and texture) was extracted from each tumor. Feature selection was employed through minimum redundancy and maximum relevance using a training set. A random forest model was developed based on top radiomic and demographic features. Model performance was evaluated by ROC analysis. SHAP plots were constructed in order to visualize feature importance in model predictions. Results: Ninety-seven patients (117 tumors, 31 (32%) microwave ablation, 66 (68%) radiofrequency ablation) were included. The mean model for end-stage liver disease (MELD) score was 10.5 ± 3. The mean follow-up time was 336.2 ± 179 days. Complete response on pathology review was achieved in 62% of patients at the time of transplant. Incomplete pathologic response was associated with four features: two first-order and two GLRM features using univariate logistic regression analysis (*p* < 0.05). The random forest model included two radiomic features (diagnostics maximum and first-order maximum) and four clinical features (pre-procedure creatinine, pre-procedure albumin, age, and gender) achieving an AUC of 0.83, a sensitivity of 82%, a specificity of 67%, a PPV of 69%, and an NPV of 80%. Conclusions: Pre-ablation MRI radiomics could act as a valuable imaging biomarker for the prediction of tumor pathologic response in patients with HCC.

## 1. Introduction

Hepatocellular carcinoma (HCC) is the sixth most common site of primary cancer and the fourth most common cause of cancer-related death worldwide. Once diagnosed, HCC management depends on several factors, including the tumor stage, patient performance status, and percentage of functional liver, therefore requiring a personalized approach. therapeutic options for HCC have evolved over the past decade; radiofrequency (RFA) or microwave ablation (MWA) therapies have emerged as an option during recent years, although surgical resection remains the best option for a potential cure for patients with early-stage HCC [1,2,3]. Tumor progression after thermal ablative therapies can lead to drop-out from the liver transplantation waiting list and decreased survival. Hence, understanding the appearance of treated lesions and response to therapy has become worthwhile in making management decisions [4,5,6]. None of the existing guidelines address the variable appearances of individual lesions after treatment, and the emphasis is more on patient-level response [7,8,9]. A treatment response algorithm was developed in the Liver Imaging Reporting and Data System due to difficulties in assessing treatment response to ablation therapies [7,8,9]. Despite the adoption of this system at many institutions, a prediction of histologic response requires extensive assessments [7,8]. Therefore, it is crucial to explore additional pragmatic methods that can be applied in order to predict the response to ablation therapy in HCC patients. There has been increasing development and application of machine learning (ML) models for predicting the presence of clinical factors from medical imaging with remarkable accuracy. Novel ML pipelines that use radiomics data and Shapley values as tools to explain outcome predictions from complex prediction models have been built in recent studies [10,11,12]. Radiomics is a method for converting images into mineable data and the subsequent data analysis for decision support [13,14,15]. Radiomics signatures specific to each patient provide valuable information for personalized medicine [16]. In a recent study, radiomics features of pre-procedural MRI were predictive of trans-arterial chemoembolization (TACE) short term treatment response in patients with HCC [17]. In this study, we assessed the role of pre-treatment tumor radiomic features in predicting pathologic response to percutaneous ablation in patients with HCC who were listed for liver transplantation.

## 2. Materials and Methods

### 2.1. Patients

This retrospective study was approved by the IRB and was compliant with the Health Insurance Portability and Accountability Act (HIPAA). Informed consent was not required. An institutional ablation database was used to identify consecutive adult patients (age ≥ 18 years) who were listed for liver transplantation with primary diagnosis of HCC and who had received either percutaneous RFA or MWA therapy as an initial bridge to transplant between January 2005 to December 2015. In addition, all patients underwent contrast-enhanced abdominal MRI within 3 months prior to ablation. The diagnosis of HCC was based on either pre-ablation percutaneous biopsy or imaging features defined by the United Network for Organ Sharing (UNOS) criteria [18]. Histopathology reports were reviewed by study investigators.

### 2.2. Data Collection

A total of 97 eligible patients (F:M 18:79, mean age 57.35 ± 7.89 years) with a combined 117 liver lesions were included. The collected clinicobiological data included age, sex, date of initial ablation, underlying cause of HCC, transplantation, serum total bilirubin levels pre- and post-ablation, pre-ablation levels of serum albumin, international normalized ratio (INR), creatinine (Cr), model for end-stage liver disease (MELD) score, occurrence of repeat ablative therapies or prior TACE therapy, and time (days) between ablation and transplantation. Histopathologic data assessed at the time of a liver explant by a board-certified surgical pathologist were collected from the electronic pathology record. Collected data included tumor measurement, histopathologic diagnosis, and tumor grade, as well as histopathologic tumor necrosis. After histopathologic review of patients who ultimately received a transplant, the complete (Responders) or incomplete (Non-responders) necrosis of ablated lesions was recorded. The main clinical, laboratory, pathology, and imaging data of the study population are summarized in Table 1.

### 2.3. Treatment Technique

All lesions (*n* = 117) were treated either with RFA (*n* = 76) or MWA (*n* = 41). All ablation procedures were performed by three fellowship-trained interventional radiologists. In RFA, an electrode produces frictional heat in water molecules in order to heat the tumor to high temperatures. Over time, fibrosis causes retraction of the necrotic tissue. A successful radiologic response is considered when the area is well-circumscribed, homogenous, and non-enhancing [19,20,21,22]. In MWA, an electrode delivers thermal energy-induced cellular destruction simultaneously to multiple tumor sites.

### 2.4. MRI Technique

MRI examinations were performed on either a 1.5 T (Signa HDx, GE Medical Systems and Magnetom Avanto, Siemens Healthcare) or 3 T (Discovery 750MR GE Medical Systems and Magnetom Trio, Siemens Healthcare) MRI system. A 32 Ch torso phased-array coil was used in all studies. Axial contrast enhanced volumetric T1-weighted fat-suppressed sequences (VIBE or LAVA) were obtained as per standard clinical protocol, including 5 mm slice thickness, 40 cm field of view, and 320 × 256 matrix [11]. Intravenous gadolinium contrast (gadopentetate/Magnevist, Bayer; gadoterate/Dotarem, Guerbet; or gadoxetate/Eovist, Bayer) was administered at the standard approved clinical dose (0.1 mmol/kg for gadopentetate and gadoterate, 0.025 mmol/kg for gadoxetate) by power injector at a rate of 1 mL/second. The arterial phase was obtained at 25–35 s post-contrast injection.

### 2.5. Tumor Segmentation

Two radiologists, blinded to all clinical information, reviewed the images to identify the hepatic lesions > 1 cm in greatest diameter by using the Picture Achieving and Communication System. Volume-based hepatic lesion segmentation was performed on DICOM arterial phase T1 post-contrast MR images (at 25–35 s post-contrast injection) by a diagnostic radiology instructor (AT, 6 years of experience in image segmentation) using an open-source volumetric image analysis software (3D slicer; http://www.slicer.org (accessed on 1 February 2021), version 4.10.2) [14,23]. An experienced fellowship-trained radiologist reviewed the segmentations (DD, 8 years of experience).

### 2.6. Extraction of Radiomic Features, Model Training and Statistical Analysis

The 112 radiomic features including first-order features, gray-level co-occurrence matrix (GLCM), gray-level dependence matrix (GLDM), gray-level run time length matrix (GLRLM), gray-level size zone matrix (GLSZM), neighboring gray-tone difference matrix (NGTDM), and 2D and 3D features, extracted from each of the 117 tumors. Figure 1 displays this method in a stepwise approach.

A random forest machine learning algorithm using a selected subset of all radiomics features was used to predict the outcome. The data were randomly partitioned into a training, validation, and testing set at a ratio of 60-20-20. Patients with more than one tumor were included in the same group to prevent overfitting. Feature selection was employed through minimum redundancy and maximum relevance (mRMR) in order to highlight top radiomic features and clinical variables in predicting disease progression. The training set was used to select features with mRMR, and each set of selected features was evaluated on the validation set. Using the training and validation sets for feature selection prevents bias in the evaluative metrics of the final model, which can result from the model obtaining data from the test set during the feature selection process. Random forest models were tested based on a combination of these features. Performance was evaluated via confusion matrix and receiver operator characteristics. In order to understand which of the selected features most strongly influenced model predictions, the SHapley Additive exPlanations (SHAP) method was applied with our model. The SHAP method is a game-theoretic approach to interpreting predictions of machine learning models and an extension of Shapley values, indicating the average marginal contribution of each feature over all combinations of features.

Univariate logistic regression identified statistically significant features in order to determine treatment response. Statistical significance was defined as a *p*-value of <0.05 for each B-coefficient on the Wald test. The machine learning and statistical analysis were performed using Python 3.8 (https://www.python.org/ (accessed on 1 November 2021)).

## 3. Results

### 3.1. Patient Characteristics

The baseline patient and clinical information prior to treatment is summarized in Table 1. Of the 97 adult patients who received radiofrequency or microwave ablation therapy, 62% of patients (*n* = 60, 14 females, mean age 57.5 ± 6.9 years) with a combined 76 (65%) liver lesions exhibited complete pathologic response on the pathology review performed at the time of transplant (Responders). The average for largest tumor size was 2.44 ± 0.67 cm, ranging 1.5–4.7 cm. Of these 60 patients, 39 (65%) patients (47/76 tumors, 62%) received RFA and 21 (35%) patients (29/76 tumors, 38%) received MWA therapy. In 14 (23.3%) Responders (17/76 tumors, 22.3%), the procedure was repeated. The most common underlying etiology in this group was HCV (36/60, 60%), followed by HBV (7/60, 11.6%), prolonged alcohol consumption (5/60, 8%), and HCV with a prolonged history of alcohol consumption (3/60, 5%). The median MELD score prior to ablation was 11. The mean total serum bilirubin levels were 1.33 ± 0.86 mg/dL and 1.54 ± 1.56 mg/dL, pre- and post-ablation, respectively.

37 patients (38%, 4 females, mean age 58.1 ± 6.2 years) showed incomplete necrosis at the time of the pathology examination performed at transplant (Non-responders). In this group, RFA and MWA were performed on 27 (29/41 tumors, 70.8%) and 10 patients (12/41 tumors, 29.2%), respectively. The procedure was repeated in eight (21.6%) patients, and at a similar rate compared to the Responders (21% vs. 23%). None of the patients underwent prior TACE therapy. The mean tumor size was 2.36 ± 0.72 cm, ranging 1–4.8 cm. The majority had underlying HCV (26/37, 70.2%), and the median pre-ablation MELD score was 10. The mean pre- and post-ablation serum bilirubin levels were 1.29 ± 0.74 mg/dL and 1.80 ± 1.36 mg/dL, respectively. The mean interval between ablation and transplantation was 19% longer in Responders vs. Non-responders (361.3 ± 187.2 vs. 295.6 ± 160.6 days).

In Figure 2, we include representative MRI images illustrating visual differences in tumor heterogeneity between a Responder who underwent RFA, and a Non-responder who underwent MWA.

### 3.2. MRI-Based Radiomics Features

Univariate logistic regression and Wald test results showed 4 features—first-order energy, first-order total energy, GLRLM long-run emphasis, and GLRLM long-run high-gray-level emphasis—to be statistically significant in determining treatment responses when used independently. The results of the univariate logistic regression and Wald tests are shown in Table 2.

### 3.3. Machine Learning for Treatment Response

A combination of radiomic and clinical features achieved the best performance, and was therefore the basis for the final random forest model. These features included diagnostics maximum, first-order maximum, pre-op creatinine, pre-op albumin, age, and gender. The original maximum is the measure of the voxel of greatest intensity in the original image. First-order statistics describe the distribution of voxel intensities in a given image. The value of the maximum is identified from a histogram of voxel intensities normalized by the number of voxels. The model achieved an AUC of 0.83 (CI: [0.67–0.96]), a sensitivity of 82%, a specificity of 67%, a PPV of 69%, and an NPV of 80%. Figure 3 displays the AUC for the final model. The combination of radiomic and clinical features outperformed each individual group. With radiomics alone, the model produced an AUC of 0.73 (CI: [0.50–0.89]), a sensitivity of 45%, a specificity of 83%, a PPV of 62%, and an NPV of 71%. With clinical features only, the model produced an AUC of 0.75 (CI: [0.57–0.91], a sensitivity of 73%, a specificity of 58%, a PPV of 62%, and an NPV of 70%. The results of all models on the testing set are shown in Table 3.

### 3.4. SHAP Plots for Feature Visualization

The SHAP plot for the final model is included in Figure 4. Features are ranked by importance in descending order: pre-procedure creatinine, age, pre-procedure albumin, first-order energy maximum, gender, and diagnostics maximum. The horizontal points vary in location based on the calculated SHAP values for each patient and whether they predict a positive or negative disease progression. The colors represent the numeric values of each variable, where red points are high and blue ones are low. For example, a low pre-procedure creatinine level was associated with increased survival.

## 4. Discussion

This study evaluated the association between MRI-based features and histopathologic tumor necrosis in order to develop a predictive machine learning model for the pathologic response to RFA or MWA therapy in patients with HCC who underwent liver transplantation, which is the optimal treatment option for patients with unresectable HCC or advanced liver disease [5,6,7]. RFA and MWA techniques destroy tumor cells locally, but complete pathologic tumor response to ablation is below the resolution of standard imaging, and new techniques are required to evaluate the residual disease. The practice of radiomics analyzes patterns of voxel signal intensity values that are reflective of microscopic lesion features and explores the underlying relationship between medical images and the phenotypic features of tumor cells [24,25]. Radiomics features capture information about the entire tumor that is difficult to obtain from a histopathological examination of biopsy specimens due to sampling error, with little additional cost and good predictive outcomes [13,26].

A number of authors previously assessed radiomics features correlated with the biological behavior and recurrence of HCC after surgical resection or in response to treatment [26,27,28,29,30,31]. Wu et al. investigated the clinical significance of MRI-based radiomics signatures for the preoperative prediction of HCC grade and reported an AUC of 0.80 for the model using radiomics combined with clinical factors [28]. Similarly, Iseke et al. developed a neural network for predicting HCC recurrence after transplant, resection, or thermal ablation, based on a combination of MRI-based radiomics and clinical variables. They achieved AUCs ranging from 0.62–0.86 for the combined model, based on post-treatment MRI in 120 cohorts [29]. A study by Liu et al. postulated a logistic regression model using T2-weighted MRI with the Barcelona Clinic liver cancer stage, and albumin-bilirubin grade was created, which produced an AUC of 0.78 in the 46-patient testing group [30]. Furthermore, authors have used radiomics extracted from contrast-enhanced CT to predict treatment response or early recurrence of HCC. Yuan et al. developed a model that produced an AUC of 0.75 when combining CT portal-venous phase radiomics with clinicopathological factors in a 55-patient testing group [26]. Sheen et al. created a multivariate logistic regression model using feature selected CT radiomics with or without tumor stage, bilobar distribution and alpha-fetoprotein levels. Radiomics and tumor stage were the most predictive of TACE responsiveness, with an AUC of 0.95 in the 20-patient testing cohort [31].

The present study suggests that the arterial phase (at 25–35 s post-contrast injection) T1 post-contrast MRI radiomics signature provides good results (AUC = 0.83) in a technically homogeneous dataset. Our random forest model consisted of two radiomic and four clinical features based on mRMR with an overall performance of 74% accuracy, 82% sensitivity, 67% specificity, 69% PPV, and 80% NPV. We demonstrated that a combination of both radiomics and clinical variables resulted in improved model performance. The present study has several strengths. First, this is one of the few large cohorts in which all patients were diagnosed with early-stage HCC, making the study population homogenous for a more robust conclusion. Second, our analysis is volumetric (3D) of the entire tumor volume, rather than a single 2D image slice. The volumetric analysis is less prone to sampling errors associated with single slice analysis and allows more complete lesion assessment. Third, the tumors were manually segmented on every slice to ensure accuracy of the extracted feature values. Fourth, patients were all followed up to the time of transplant and underwent transplantation, allowing for a true evaluation of pathologic response. Additionally, the inclusion of SHAP plots highlighted feature importance and enhanced model transparency.

Despite the novelties of our study, there are some limitations. Since it is a single-center study, it lacks external validation; thus, the reproducibility and generalizability of our findings remain to be verified. In addition, all included patients underwent pre-ablation MRI. This reduced heterogeneity but limited the generalizability of our results to patients imaged with other imaging modalities. Lastly, the diagnosis of HCC was based on either pre-ablation percutaneous biopsy or imaging features defined by the UNOS criteria.

## 5. Conclusions

The machine-learning-based radiomics features analysis could non-invasively explore the association between pre-ablation MRI images and pathological response, and provide a prediction model in patients with HCC who were listed for hepatic transplantation. This would allow us to predict who will benefit from ablation therapy, and personalize treatment for these patients. A larger dataset and external validation would be required in order to truly establish a model like this in future.

## Figures and Tables

**Figure 1 cancers-15-02058-f001:**
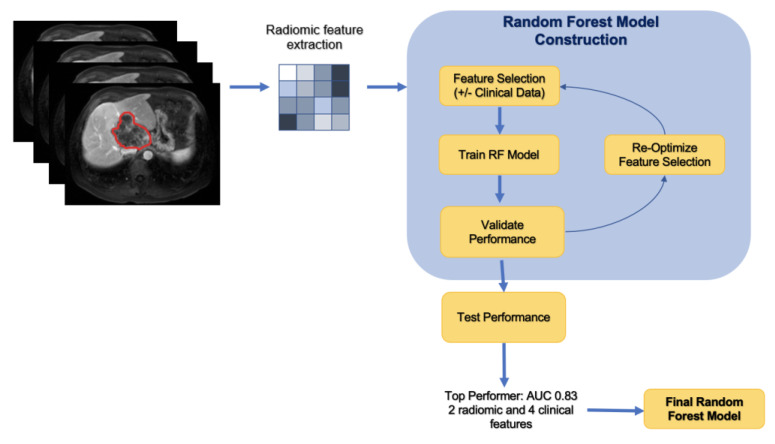
Stepwise approach of the machine learning model development.

**Figure 2 cancers-15-02058-f002:**
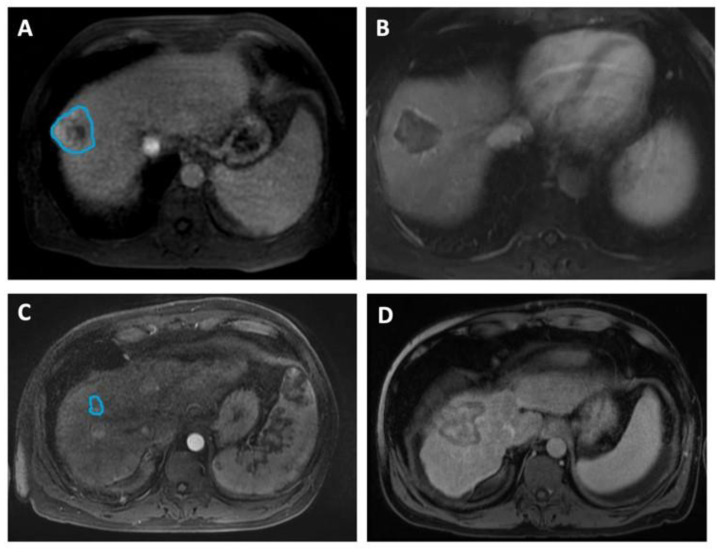
Four representative images illustrating differences in HCC tumor heterogeneity between two patients who underwent different ablation therapies and had contrasting treatment outcomes. (**A**,**B**) Axial MRI images of a 58-year-old male demonstrate (**A**) a 4.6 cm mass (manually delineated; blue ROI) in the right hepatic lobe, 5 days before RFA and (**B**) a 4.2 cm zone of ablation 50 days prior to the liver transplant. He remained recurrence-free during 1139 days of follow-up, up to the time of the transplant. The histopathologic review at the time of transplant reported complete necrosis. (**C**,**D**) Axial MRI images of a 49-year-old male illustrate (**C**) a 1.6 cm lesion (manually delineated; blue ROI) in the right hepatic lobe, 90 days prior to MWA and (**D**) a 5.1 cm ablation zone 5 days before the transplant. The patient underwent repeated procedures during the 366 days of follow-up, and exhibited incomplete pathologic response at the time of transplant.

**Figure 3 cancers-15-02058-f003:**
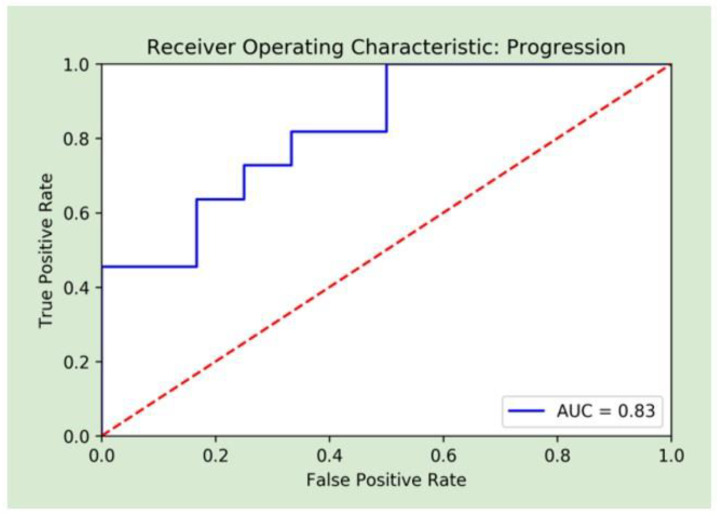
Receiver operative curve analysis of the random forest combined machine learning model.

**Figure 4 cancers-15-02058-f004:**
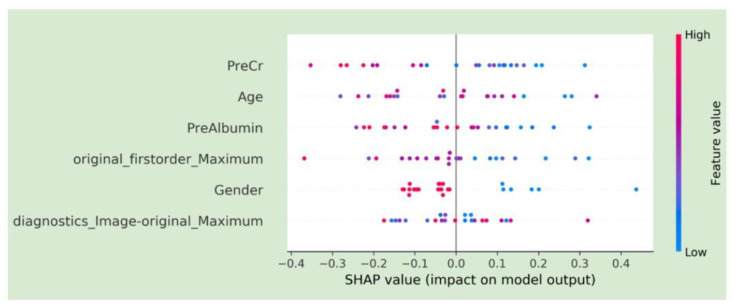
SHAP plots demonstrating feature characteristics and importance for model predictions.

**Table 1 cancers-15-02058-t001:** Patient characteristics prior to treatment.

	Value
**No. of subjects**	97
**Mean age ^a^ (years ± SD)**	57.3 ± 7.8
**Sex (F:M)**	18:79
**Interval between ablation and transplantation (d)**	336.22 ± 179.60
**Total bilirubin level (mg/dL)**	1.31 ± 0.81
**Albumin level**	3.65 ± 0.68
**INR level**	1.24 ± 0.16
**Creatinine level**	0.89 ± 0.22
**MELD score**	
Mean	10.56 ± 3.4
Median	11
Range	6–21
**Underlying cause of liver disease ***	
Hepatitis C only	60
Alcohol only	10
Hepatitis B only	9
Hepatitis C and Alcohol	5
Nonalcoholic Steatohepatitis	4
Nonalcoholic Fatty Liver Disease	2
Hemochromatosis	2
Hepatitis B and C	1
Primary Biliary Cholangitis	1
Autoimmune Hepatitis	1
Budd-Chiari	1
Cryptogenic	1
**Pre-embolization lesion size (mm)**	2.41 ± 0.69
**Treatment modality**	
Percutaneous Microwave ablation	31
Percutaneous Radiofrequency ablation	66
Repeat Procedure	22

^a^ At the time of MRI. * The cause of liver disease that led to hepatic transplant.

**Table 2 cancers-15-02058-t002:** Univariate logistic regression analysis for predicting treatment response in the radiomics-only model. Only features with statistically significant Wald test results are shown.

Features	Coefficient (B)	95% Confidence Interval	*p*-Value
**First−order_Energy**	−3.98 × 10^−5^	(−9.27 × 10^−6^ to 7.04 × 10^−5^)	0.011
**First−order_TotalEnergy**	−4.80 × 10^−5^	(−1.12 × 10^−5^ to 8.47 × 10^−5^)	0.011
**Glrlm_LongRunEmphasis**	−5.4 × 10^−3^	(−1.98 × 10^−4^ to −1.06 × 10^−2^)	0.042
**Glrlm_LongRunHighGrayLevelEmphasis**	−5.9 × 10^−3^	(−9.92 × 10^−4^ to −1.08 × 10^−2^)	0.019

**Table 3 cancers-15-02058-t003:** Performance metrics of all models on the testing set.

Model	AUC	Sensitivity	Specificity	PPV	NPV
Clinical features only	0.75	0.73	0.58	0.62	0.70
Radiomics only	0.73	0.45	0.83	0.62	0.71
Combined Radiomics and clinical features	0.83	0.82	0.67	0.69	0.80

## Data Availability

The data presented in this study are available on request from the corresponding author.

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
