# Peer review of "Machine Learning-Based Radiomic Features on Pre-Ablation MRI as Predictors of Pathologic Response in Patients with Hepatocellular Carcinoma Who Underwent Hepatic Transplant"

_cancers, 2023, doi:10.3390/cancers15072058_

Round 1
Reviewer 1 Report
1. When only one sequence is used for prediction, the amount of information available is often limited. Have other sequences, such as DWI and T1, been considered for inclusion in the analysis?
2. Given the limited size of the patient cohort, it is advisable to employ a cross-validation strategy with multiple folds. However, it appears that the article only utilized a single fold.
3. The article's description of the extraction of radiomic features is somewhat unclear, particularly with regard to the tools and filtering operators used for feature extraction. Could you clarify the specific tools and filtering operators used in the extraction process?
4. Was there any selection process applied to the clinical features in the study? The article does not provide a specific description of such a process.
5. The statistical analysis of results for individual clinical features, individual radiomic features, and combined features appears incomplete. It would be advisable to include additional visual aids or a separate table to supplement the current analysis. Additionally, some statistical measures, such as confidence intervals, have not been accounted for in the analysis.
6. There are some errors in the article, including typos and mistakes in the figures, that require attention. For instance, in line 95, "RF" should be corrected to "RFA", and in Figure 2, there appears to be a labeling error in panel A with the inclusion of a "B". The authors are encouraged to thoroughly review the manuscript to identify and address such issues.
Author Response
Dear Reviewer,
Thank you for reviewing our manuscript ID: cancers-2252798 entitled "Machine Learning-based Radiomic Features on Pre-Ablation MRI as Predictors of Pathologic Response in Patients with Hepatocellular Carcinoma underwent Hepatic Transplant” and for inviting us to submit a revised manuscript. We have modified the paper according to the Reviewer comments. Our point-by-point response is below. Thank you again for considering this revised manuscript and we hope it is deemed worthy of publication in Cancers Journal.
Best regards,
Azadeh Tabari, MD
Division of Interventional Radiology, Department of Radiology
Massachusetts General Hospital, Harvard Medical School, Boston, MA
Reviewer1
- When only one sequence is used for prediction, the amount of information available is often limited. Have other sequences, such as DWI and T1, been considered for inclusion in the analysis?
We thank the reviewer for this question. For consistency, we have only included arterial phase T1 post-contrast MR images (at 25-35 seconds post-contrast injection) to perform volume-based hepatic lesion segmentation. Therefore, no other sequences were considered.
- Given the limited size of the patient cohort, it is advisable to employ a cross-validation strategy with multiple folds. However, it appears that the article only utilized a single fold.
We thank the reviewer for this observation. Given the sample size of the study, our goal was to develop a model using radiomics +/- clinical features that could eventually be applied to a larger study. We believe the 60-20-20 training-validation-testing set in combination with SHAP plots demonstrates the generalizability and future impact of this model based on current literature. We re-optimized the model based on the validation set.
- The article's description of the extraction of radiomic features is somewhat unclear, particularly with regard to the tools and filtering operators used for feature extraction. Could you clarify the specific tools and filtering operators used in the extraction process?
We agree with the reviewer and have revised section 2.6. in the manuscript.
- Was there any selection process applied to the clinical features in the study? The article does not provide a specific description of such a process.
We thank the reviewer for this comment. We were limited by the clinical features available in the database. However, we extracted the relevant clinical features for liver disease and HCC.
“The collected clinicobiological data included age, sex, dates of initial ablation, underlying cause of HCC, transplantation, serum total bilirubin levels pre- and post-ablation, pre-ablation levels of serum albumin, international normalized ratio (INR), creatinine (Cr), model for end-stage liver disease (MELD) score, occurrence of repeat ablative therapies or prior TACE therapy and time (days) between ablation and transplantation. Histopathologic data assessed at the time of a liver explant by a board-certified surgical pathologist were collected from the electronic pathology record.”
- The statistical analysis of results for individual clinical features, individual radiomic features, and combined features appears incomplete. It would be advisable to include additional visual aids or a separate table to supplement the current analysis. Additionally, some statistical measures, such as confidence intervals, have not been accounted for in the analysis.
We thank the reviewer for this comment. The model uses advanced computer architecture to combine these features to predict outcomes that are not applicable to certain statistical tests. Because of this, we included the SHAP plots to aid in model transparency. These plots are now being commonly used in literature for that purpose. These plots show the individual strength and value of each feature in the complex outcome prediction. For example in Figure 4, under “age” the blue dots to the right of the vertical line (0.0) represent younger age and increased survival. Similarly, under “original_firstorder_Maximum” low values (blue dots) lead to increased survival and the purple dots are of medium value but are more associated with decreased survival. We have also added confidence intervals for AUC.
- There are some errors in the article, including typos and mistakes in the figures, that require attention. For instance, in line 95, "RF" should be corrected to "RFA", and in Figure 2, there appears to be a labeling error in panel A with the inclusion of a "B". The authors are encouraged to thoroughly review the manuscript to identify and address such issues.
Thank you. The appropriate changes have been made.

Reviewer 2 Report
comment 1 (abstract)
What do authors mean by "and demographic features"?
comment 2 (abstract)
The authors state "model included two radiomic features (diagnostics-maximum and first order-maximum) and four 37 clinical features (pre-procedure creatinine, pre-procedure albumin, age and gender)".
Does 'diagnostics' features used as input features to train the model? Explain and clarify this aspect. (I have reviewed several radiomics papers and this is the first time I have seen 'diagnostics' used).
comment 3
Having already calculated sensitivity and specificity, it is very easy to calculate PPV and NPV, which in clinical work is always desirable to have.
comment 4 (1. Introduction)
The section doesn't report advantages deriving from the use of intepretable features and explainable predictive models. In order to provide a comprehensive overview, the following literature works should be added and discussed.
- Severn, C., Suresh, K., Görg, C., Choi, Y. S., Jain, R., & Ghosh, D. (2022). A Pipeline for the Implementation and Visualization of Explainable Machine Learning for Medical Imaging Using Radiomics Features. Sensors, 22(14), 5205. https://doi.org/10.3390/s22145205
- Chetoui, M., & Akhloufi, M. A. (2022). Explainable vision transformers and radiomics for covid-19 detection in chest x-rays. Journal of Clinical Medicine, 11(11), 3013. https://doi.org/10.3390/jcm11113013
- Militello, C., Prinzi, F., Sollami, G., Rundo, L., La Grutta, L., & Vitabile, S. (2023). CT Radiomic Features and Clinical Biomarkers for Predicting Coronary Artery Disease. Cognitive Computation, 1-16. https://doi.org/10.1007/s12559-023-10118-7
comment 5 (2.2. Data Collection)
The authors state "Ninety-seven eligible patients (F: M 18:79, mean age 57.35 ± 7.89 years) with 117 liver 81 lesions were included". This means that for some patients more than one tumour has been considered.
Please, provide further details about. Moreover, the correlation among tumours belonging to the same patient was considered in the training of the model.
comment 6 (2.6. Machine Learning and Statistical Analysis)
I suggest renaming the section "Model Training and Statistical Analysis"
comment 7 (2.6. Machine Learning and Statistical Analysis)
What kind of procedure was used to train the model? A cross-validation?
comment 8 (3.2. MRI-based Radiomics Features)
In table 2 only statistically significant features are reported. Was statistical significance assessed upstream of the modelling? How come logistic regression is mentioned, and how much then the predictive model uses random forest between the two radiomic features considered (first-order energy-maximum, diagnostics-maximum) none of them are present in table 2. The discussion is unclear and even seems confusing in some passages.
Review the entire section 3 and clarify the various passages.
comment 9 (3. Results)
A table summarizing all the obtained metrics (specificity, sensitivity, AUC,....) must be added, reporting averaged values and standard deviation.
Author Response
Dear Reviewer,
Thank you for reviewing our manuscript ID: cancers-2252798 entitled "Machine Learning-based Radiomic Features on Pre-Ablation MRI as Predictors of Pathologic Response in Patients with Hepatocellular Carcinoma underwent Hepatic Transplant” and for inviting us to submit a revised manuscript. We have modified the paper according to the Reviewer comments. Our point-by-point response is below. Thank you again for considering this revised manuscript and we hope it is deemed worthy of publication in Cancers Journal.
Best regards,
Azadeh Tabari, MD
Division of Interventional Radiology, Department of Radiology
Massachusetts General Hospital, Harvard Medical School, Boston, MA
Reviewer 2
comment 1 (abstract)
What do authors mean by "and demographic features"?
We thank the reviewer for pointing out this typo. We have edited the abstract.
comment 2 (abstract)
The authors state "model included two radiomic features (diagnostics-maximum and first order-maximum) and four 37 clinical features (pre-procedure creatinine, pre-procedure albumin, age and gender)".
Does 'diagnostics' features used as input features to train the model? Explain and clarify this aspect. (I have reviewed several radiomics papers and this is the first time I have seen 'diagnostics' used).
Thanks for your question. “Diagnostic image maximum” is a first-order characteristics, a radiomics feature described in Python package 3.8 (https://www.python.org/).
comment 3
Having already calculated sensitivity and specificity, it is very easy to calculate PPV and NPV, which in clinical work is always desirable to have.
We thank the review for this suggestion. We have added PPV and NPV to the results.
comment 4 (1. Introduction)
The section doesn't report advantages deriving from the use of intepretable features and explainable predictive models. In order to provide a comprehensive overview, the following literature works should be added and discussed.
- Severn, C., Suresh, K., Görg, C., Choi, Y. S., Jain, R., & Ghosh, D. (2022). A Pipeline for the Implementation and Visualization of Explainable Machine Learning for Medical Imaging Using Radiomics Features. Sensors, 22(14), 5205. https://doi.org/10.3390/s22145205
- Chetoui, M., & Akhloufi, M. A. (2022). Explainable vision transformers and radiomics for covid-19 detection in chest x-rays. Journal of Clinical Medicine, 11(11), 3013. https://doi.org/10.3390/jcm11113013
- Militello, C., Prinzi, F., Sollami, G., Rundo, L., La Grutta, L., & Vitabile, S. (2023). CT Radiomic Features and Clinical Biomarkers for Predicting Coronary Artery Disease. Cognitive Computation, 1-16. https://doi.org/10.1007/s12559-023-10118-7
We thank the reviewer for the suggestions and have added a paragraph to introduction. These references are cited in the manuscript.
comment 5 (2.2. Data Collection)
The authors state "Ninety-seven eligible patients (F: M 18:79, mean age 57.35 ± 7.89 years) with 117 liver 81 lesions were included". This means that for some patients more than one tumour has been considered.
Please, provide further details about. Moreover, the correlation among tumours belonging to the same patient was considered in the training of the model.
Thank you for pointing this out. 14/97 patients had >1 liver tumors (10 patients with 2 liver tumors, 2 patients with 3 tumors and 2 patients with 4 tumors). All tumors that belong to a patient was either in the testing or training datasets.
comment 6 (2.6. Machine Learning and Statistical Analysis)
I suggest renaming the section "Model Training and Statistical Analysis"
Thank you for the suggestion. We have renamed the section.
comment 7 (2.6. Machine Learning and Statistical Analysis)
What kind of procedure was used to train the model? A cross-validation?
We used a train-validation-test model that was optimized based on the validation set and final results based on the testing set. We have also illustrated this is in Figure 1. Stepwise approach of the machine learning model development.
comment 8 (3.2. MRI-based Radiomics Features)
In table 2 only statistically significant features are reported. Was statistical significance assessed upstream of the modelling? How come logistic regression is mentioned, and how much then the predictive model uses random forest between the two radiomic features considered (first-order energy-maximum, diagnostics-maximum) none of them are present in table 2. The discussion is unclear and even seems confusing in some passages.
Review the entire section 3 and clarify the various passages.
Thank you for raising this important point of clarification. The univariate logistic regression was not a component of the machine learning model. It was used to show significance to outcome predictions based on each radiomic feature independently. The 4 statistically significant features were not included in the final machine learning analysis. This demonstrates the strength of feature integration rather than single extraction.
comment 9 (3. Results)
A table summarizing all the obtained metrics (specificity, sensitivity, AUC,....) must be added, reporting averaged values and standard deviation.
Thank you for your comment. Table 3 summarized the performance metrics of all models.
Table 3. Performance metrics of all models on the testing set
|
Model |
AUC |
Sensitivity |
Specificity |
PPV |
NPV |
|
Clinical features only |
0.75 |
0.73 |
0.58 |
0.62 |
0.70 |
|
Radiomics only |
0.73 |
0.45 |
0.83 |
0.62 |
0.71 |
|
Combined Radiomics and clinical features |
0.83 |
0.82 |
0.67 |
0.69 |
0.80 |

Round 2
Reviewer 2 Report
All concerns raised in the previoius manuscript version have been properly approached.
Author Response
Thank you